# Targeting Ca^2+^ Signaling in the Initiation, Promotion and Progression of Hepatocellular Carcinoma

**DOI:** 10.3390/cancers12102755

**Published:** 2020-09-24

**Authors:** Eunus S. Ali, Grigori Y. Rychkov, Greg J. Barritt

**Affiliations:** 1Department of Medical Biochemistry, College of Medicine and Public Health, Flinders University, Adelaide 5001, South Australia, Australia; eunus.ali_csl@yahoo.com.au; 2School of Medicine, The University of Adelaide, Adelaide 5005, South Australia, Australia; grigori.rychkov@adelaide.edu.au; 3South Australian Health and Medical Research Institute, Adelaide 5005, South Australia, Australia

**Keywords:** liver, Ca^2+^, hepatocellular carcinoma, cancer stem cells, reactive oxygen species, mitochondria, STIM1, TRP channels

## Abstract

**Simple Summary:**

Liver cancer (hepatocellular carcinoma) is a significant health burden worldwide. It is often not detected until at an advanced stage when there are few treatment options available. Changes in calcium concentrations within liver cancer cells are essential for regulating their growth, death, and migration (metastasis). Our aim was to review published papers which have identified proteins involved in calcium signaling as potential drug targets for the treatment of liver cancer. About twenty calcium signaling proteins were identified, including those involved in regulating calcium concentrations in the cytoplasm, endoplasmic reticulum and mitochondria. A few of these have turned out to be sites of action of natural products previously known to inhibit liver cancer. More systematic studies are now needed to determine which calcium signaling proteins might be used clinically for treatment of liver cancer, especially advanced stage cancers and those resistant to inhibition by current drugs.

**Abstract:**

Hepatocellular carcinoma (HCC) is a considerable health burden worldwide and a major contributor to cancer-related deaths. HCC is often not noticed until at an advanced stage where treatment options are limited and current systemic drugs can usually only prolong survival for a short time. Understanding the biology and pathology of HCC is a challenge, due to the cellular and anatomic complexities of the liver. While not yet fully understood, liver cancer stem cells play a central role in the initiation and progression of HCC and in resistance to drugs. There are approximately twenty Ca^2+^-signaling proteins identified as potential targets for therapeutic treatment at different stages of HCC. These potential targets include inhibition of the self-renewal properties of liver cancer stem cells; HCC initiation and promotion by hepatitis B and C and non-alcoholic fatty liver disease (principally involving reduction of reactive oxygen species); and cell proliferation, tumor growth, migration and metastasis. A few of these Ca^2+^-signaling pathways have been identified as targets for natural products previously known to reduce HCC. Promising Ca^2+^-signaling targets include voltage-operated Ca^2+^ channel proteins (liver cancer stem cells), inositol trisphosphate receptors, store-operated Ca^2+^ entry, TRP channels, sarco/endoplasmic reticulum (Ca^2+^+Mg^2+^) ATP-ase and Ca^2+^/calmodulin-dependent protein kinases. However, none of these Ca^2+^-signaling targets has been seriously studied any further than laboratory research experiments. The future application of more systematic studies, including genomics, gene expression (RNA-seq), and improved knowledge of the fundamental biology and pathology of HCC will likely reveal new Ca^2+^-signaling protein targets and consolidate priorities for those already identified.

## 1. Introduction

Hepatocellular carcinoma (HCC)—which arises from chronic liver disease—is a significant contributor to cancer-related deaths and often has poor clinical outcomes. Due to the prevalence of obesity, non-alcoholic fatty liver disease (NAFLD), and metabolic syndrome the incidence of HCC is increasing [1,2,3]. The liver is a complex organ, composed of several different cell types, of which hepatocytes form the majority [4]. Hepatocytes are responsible for the central metabolism of the body including carbohydrate, lipid, protein and amino acid metabolism. Hepatocytes also mediate the uptake, synthesis and export of bile acids and the metabolism and excretion of drugs and other xenobiotic molecules [4]. Changes in intracellular free Ca^2+^ concentrations play a central role in the hormonal regulation of liver metabolism and of the pathways which regulate hepatocyte proliferation and apoptosis [5]. Factors that create an environment in the liver which initiate and promote HCC include hepatitis B (HBV), hepatitis C (HCV) and liver steatosis [1,2,3]. Liver cancer stem cells are thought to play an important role in the initiation and progression of HCC [6,7,8,9,10].

Numerous mutations in genes encoding Ca^2+^-signaling proteins have been identified in DNA extracted from HCC liver tissue, and a number of Ca^2+^-signaling proteins are over or under expressed in HCC [11,12]. These and many other studies conducted over the past 20 years have identified Ca^2+^-signaling proteins which appear to play important roles on the initiation and progression of HCC and which could be potential therapeutic targets. These potential targets are summarized in Table 1. The table was arranged so as to group the potential Ca^2+^-signaling protein targets in the different stages of HCC: liver cancer stem cells, initiation and promotion by HBV and HCV infection and non-alcoholic fatty liver disease, and in established HCC tumors, including migration and metastasis. The aim of this review is to identify what is known about these potential Ca^2+^-signaling proteins, to evaluate the evidence for this identification and explain the rationale for targeting these steps in the treatment of HCC.

Liver anatomy and biology, and the pathways of initiation and progression of HCC are complex. Therefore, the first part of this review sets out the biologic and pathologic background which forms the context for the identification and evaluation of potential Ca^2+^-signaling targets in HCC. Of particular importance are the roles of liver cancer stem cells in HCC initiation, progression and resistance to treatments. Also, the potential of drug-emitting bead transcatheter arterial chemoembolization to deliver a drug to the site of a tumor. Many studies have followed a general pattern. This has often involved initially identifying a given Ca^2+^-signaling protein or pathway of potential interest. Then experiments have been conducted to measure the expression of the Ca^2+^-signaling protein(s) in human HCC and non-tumor liver tissue and in HCC cell lines and to artificially alter expression of the protein in HCC cell lines and examine the effects on cell proliferation, apoptosis, and/or metastasis. Finally, downstream pathways potentially involved are often investigated. In other studies, bioinformatics analysis or other ideas have led to the identification of a protein which has turned out to be a Ca^2+^-signaling protein.

## 2. Overview of Liver Anatomy, Cell Types, Physiology and Metabolic Pathways

The metabolic roles of the liver include the metabolism of carbohydrates, lipids, proteins and amino acids and the synthesis and transport of the components of bile fluid and the metabolism and excretion via bile or blood of drugs and xenobiotic compounds [4,5,40]. These processes are principally conducted by hepatocytes (parenchymal cells). Several other cell types are also present in the liver. These include cholangiocytes, (which are also parenchymal cells) and fibroblasts, stellate cells, Kupffer cells and sinusoidal endothelial cells (which are all nonparenchymal cells) [4,5,40]. In addition, liver tissue harbors liver precursor/stem cells, as discussed in more detail below. Having an appreciation that all these different cell types are likely to be present in any given sample of liver tissue is an important aspect of interpreting both Ca^2+^-signaling pathways and changes in signaling proteins in HCC tissue.

Hepatocytes constitute 60–80% of the total liver mass. They are normally quite stable with an average lifetime (turnover) of many days. However, hepatocytes have a large capacity to proliferate. This allows the liver to regenerate following liver surgery [4,5,40]. About 30% of hepatocytes are polyploid, which means they contain at least two sets of homologous chromosomes. This is due to both an increase in the amount of DNA in a given nucleus and/or to an increase in the number of nuclei per cell [41,42].

The smallest defined subanatomical unit of the liver is the hexagonal lobule. In the human liver this has a diameter of about 1 mm. A portal triad is located at each of the six corners of the hexagon. This consists of a portal artery, portal vein and bile duct, arranged as shown schematically in Figure 1 [4]. This figure also shows the direction of blood and bile flow and the relative anatomic locations of hepatocytes, stellate, sinusoidal endothelial and Kupffer cells. To achieve their functions, hepatocytes are spatially polarized with sinusoidal, canalicular and basolateral membranes (Figure 1). Cholangiocytes line the bile duct and are responsible for the movement of bile fluid out of the liver to the common bile duct. Kupffer cells are resident macrophages and are responsible for protecting the liver from inflammation. Functions of stellate cells include mediating the development of liver fibrosis following liver inflammation and/or injury.

## 3. Role of Intracellular Ca^2+^ in Regulation of Hepatocyte Metabolism, Proliferation, Injury and Death

Pathways of intracellular Ca^2+^-signaling in hepatocytes have been well characterized over the past 30 to 40 years (reviewed in [5]). Hepatocytes were, in part, a model cell type for understanding Ca^2+^-signaling in non-excitable cells. These signaling pathways included the role of Ca^2+^ in the hormonal regulation of glycolysis and one of the first observations of Ca^2+^ oscillations in animal cells [43,44]. Likewise, intracellular Ca^2+^-signaling in cholangiocytes was well characterized (reviewed in [45,46]). However, much less is known about intracellular Ca^2+^-signaling in Kupffer, stellate and sinusoidal endothelial cells. Yet, as indicated above, these cell types play important roles in liver function. They are involved in the initiation and progression of HCC and are present in samples of HCC liver and non-tumor liver.

It is worthwhile considering why targeting Ca^2+^-signaling proteins, rather than targeting protein kinases (for example), may be a useful strategy in the prevention and treatment of HCC. Intracellular Ca^2+^ is a widespread and reversible signaling system in animal cells. At an early stage in the evolution of living cells, formation of the cell membrane was also associated with the creation of a very large Ca^2+^ gradient (about 20,000) across the cell membrane. This evolved cell membrane arrangement protected the cytoplasm and organelles within the cell from the high Ca^2+^ concentrations in the external environment [47,48]. Enclosure and definition of the intracellular space were coupled with the evolution of many proteins possessing the ability to bind Ca^2+^ at Ca^2+^-specific binding sites. This allowed Ca^2+^-signaling to impinge on most biochemical pathways [47,48]. On one hand, this makes Ca^2+^-signaling an attractive pharmaceutical target, as the inhibition or activation of a given process can have multiple downstream consequences which may potentially be beneficial. On the other hand, the broad nature of the Ca^2+^-signaling system means that interaction at one site has the potential to affect numerous downstream events and hence may not be very specific. Choosing Ca^2+^-signaling targets for the prevention and treatment of HCC may thus involve a delicate balance between these two extremes.

In adult hepatocytes, changes in the free Ca^2+^ concentration in the cytoplasmic space ([Ca^2+^]_cyt_), mitochondrial matrix ([Ca^2+^]_MT_), lumen of the endoplasmic reticulum ([Ca^2+^]_ER_), nucleus, and lysosomes play central roles in the hormonal regulation of glycogen metabolism, glycolysis, lipid synthesis and degradation, and protein and amino acid metabolism (reviewed in [5]). Intracellular Ca^2+^-signaling is also essential for regulating the transport and movement of the components of bile fluid. Of particular importance in the development, progression and therapeutic treatment of HCC are the roles of intracellular Ca^2+^ in regulating the proliferation, apoptosis and migration of hepatocytes and HCC cells. Some key pathways are shown in Figure 2 and Figure 3. Together with other signaling pathways, changes in intracellular Ca^2+^ can switch cell signaling from proliferation to apoptosis and cell death. Moreover, interventions causing a large influx of Ca^2+^ across the plasma membrane, such as may arise from the pathologic or pharmacological activation of some TRP channels, also induces cell death [31,32,49].

The “Ca^2+^-signaling tool-kit” in the adult hepatocyte is similar to that in many other non-excitable mammalian cells (reviewed in [5]). However, there are some differences in the expression of channels and transporters between adult hepatocytes and those in progenitor/stem cells, liver cancer stem cells and HCC cells. Moreover, in cholangiocytes, Kupffer cells, stellate and sinusoidal epithelial cells different aspects of Ca^2+^-signaling pathways are emphasized. As discussed below, one of these differences is the expression of voltage-operated Ca^2+^ channels which do not seem to be expressed in adult hepatocytes but may be expressed in other cells present in HCC tumors.

Most studies of Ca^2+^-signaling in hepatocytes have been conducted using isolated rat or mouse hepatocytes, liver cell lines established from rat or mouse liver tumors and in liver cell lines established from human liver tumors. Ca^2+^-signaling proteins and pathways in normal human hepatocytes are less well characterized. However, most of the studies reported in this review involve human liver tissue and liver cell lines derived from human liver. While the Ca^2+^-signaling pathways in normal human adult hepatocytes are likely to be broadly similar to those in normal rat and mouse adult hepatocytes, there are some differences.

The three sub-types of InsP_3_ receptors (InsP_3_R) are expressed in adult rat and mouse hepatocytes, although the level of expression of type 3 InsP_3_R (InsP_3_R3) is low. The roles of type 2 InsP_3_ receptors (InsP_3_R2) in generating oscillations in [Ca^2+^]_cyt_ and in the regulation of bile secretion and Ca^2+^-signaling in the hepatocyte nucleus have been well characterized [45,50]. Type 1 InsP_3_ receptors (InsP_3_R1) are involved in the regulation of metabolism and may be specifically associated with the activation of Ca^2+^ entry (reviewed in [5,45]). It is unlikely that there are functional ryanodine receptors in adult rat and mouse hepatocytes, but this point is not fully resolved (reviewed in [5]).

In rat and mouse hepatocytes, the main Ca^2+^ entry pathway is via store-operated Ca^2+^ entry, mediated by Orai and STIM polypeptides [5]. These permit sustained hormone-induced oscillations in [Ca^2+^]_cyt_ and maintain adequate Ca^2+^ in the ER lumen during hormonal Ca^2+^-signaling. Thus, Ca^2+^ entry through store-operated Ca^2+^ channels replenishes intracellular Ca^2+^ which is extruded from the cytoplasmic space via the plasma membrane (Ca^2+^+Mg^2+^)ATP-ase following an increase in [Ca^2+^]_cyt_. In hepatocytes isolated from human liver, TRPC6 polypeptides may also make a contribution to store-operated Ca^2+^ entry [28]. Moreover, several other TRP non-selective cation channels are also expressed in hepatocytes. These likely serve to deliver Ca^2+^ and Na^+^ to the cytoplasmic space to serve specific regulatory functions [5].

Store-operated Ca^2+^ entry to hepatocytes and liver cell lines is frequently measured using an intracellular Ca^2+^ sensor (e.g., Fura-2) and fluorescence imaging employing a “Ca^2+^ add back” protocol. This involves measurement of the increase in [Ca^2+^]_cyt_ following the addition of extracellular Ca^2+^ to the culture medium of cells previously incubated in the absence of extracellular Ca^2+^ plus an agent such as thapsigargin or a generator of InsP_3_ to release Ca^2+^ from the ER. While this is a useful and valuable technique, measurement of Ca^2+^ entry by patch clamp recording is needed to confirm the presence of functional store-operated Ca^2+^ channels. These are normally defined as possessing the properties of Ca^2+^-release-activated Ca^2+^ channels (CRAC) channels, originally identified in lymphocytes and mast cells [51].

To our knowledge, there is no evidence for the expression of functional voltage-operated Ca^2+^ channels in hepatocytes isolated from normal adult rat, mouse or human liver [5]. However, as discussed below, voltage-operated Ca^2+^ channels are likely expressed in liver cancer stem cells, in HCC cells and in immortalized HCC cell lines (reviewed in [5]). Moreover, stellate and Kupffer cells also express voltage-operated Ca^2+^ channels (reviewed in [5]).

## 4. The Pathology of Hepatocellular Carcinoma

In the context of all cancers, HCC poses a significant health burden and is a major cause of cancer-related deaths in both men and women [3,52,53,54,55]. With the current increased prevalence of obesity, non-alcoholic fatty liver disease and metabolic syndrome, the incidences of HCC and HCC-related deaths are increasing [56]. HCC is one of two main primary liver cancers (i.e., cancers that originate from mutations in hepatocytes or other cell types in the liver) and accounts for about 80% of all liver cancers [3,10]. The other main primary liver cancer is intrahepatic cholangiocarcinoma, which accounts for about 1% of all liver cancers. Other primary liver cancers are mixed hepatocellular cholangiocarcinoma, fibrolamellar hepatocellular carcinoma and pediatric hepatoblastoma. Metastatic cancers (i.e., non-primary liver cancers), the most important of which is colon cancer, can also be established in the liver [57]. Although these non-primary liver cancers are also important, in this review we only consider HCC.

HBV and HCV were previously the major risk factors for HCC [53,54]. However, hepatitis viral infections can now be well treated and controlled, so that non-alcoholic fatty liver disease is considered to be the leading risk factor for HCC. Other risk factors include aflatoxins and type 2 diabetes [53,58].

Stages in the development of HCC from initiation to metastasis to other regions of the liver and eventually to other tissues are summarized schematically in Figure 4 [3,10,59]. The development of HCC tumors is usually preceded by liver fibrosis and cirrhosis, although some HCCs develop without cirrhosis [1,3,10,60]. An example of an HCC tumor (being removed surgically by liver resection) and an example of the histology of liver tissue across a tumor, the tumor margin and para tumor tissue are shown in Figure 5. The transformation of HCC cells to acquire migratory and metastatic properties involves a change from an epithelial to a mesenchymal phenotype. In this transition, epithelial cells lose spatial cell polarity and cell–cell adhesiveness and gain migratory and invasive properties [3].

In the early stages, HCC does not normally give rise to many physical symptoms and signs. Early stage HCC can usually only be detected using ultrasound, imaging and measurement of blood alpha-fetoprotein concentrations. In the detection and monitoring of later stages of HCC, imaging and blood alpha-fetoprotein play major roles [10,53].

The mechanisms involved in the initiation and progression of HCC are complex and are only partly understood. Epigenetic as well as genetic changes are involved. Mutated genes which feature in many HCCs include those encoding proteins which regulate the Wnt/β-catenin pathway, the p53 cell cycle pathway, telomere maintenance and chromatin structure and function [10,11,60,62,63]. As discussed below, stem cells are thought to play an important role in the initiation and progression of HCC [6,7,8,9,10,60]. Development and progression of HCC is promoted by inflammation, such as that initiated by HBV and HCV and steatosis (non-alcoholic fatty liver disease) [53,64].

## 5. Current Treatments for Hepatocellular Carcinoma

Current treatment options for HCC at the different stages are summarized in Figure 4. Well established HCC is difficult to treat, resulting in uncertain and often poor outcomes [3,65,66]. If HCC is detected in the very early stages with only one, or a few, tumor nodules of small size, the tumor(s) can be removed surgically by liver resection or liver transplantation (surgical liver resection shown in Figure 5A). Examples of systemic agents used to treat later stage HCC include sorafenib and lenvatinib (multikinase inhibitors), PD-L1 (programmed death-ligand 1) receptor blockers, statins and metformin [3]. Unfortunately, for many treatments the risk of cancer recurrence is high.

Of particular interest in considering the potential administration of therapeutic agents targeted to Ca^2+^-signaling pathways in HCC is drug-emitting bead transcatheter arterial chemoembolization. This is employed to deliver therapeutic agents to the site of tumors in the treatment of HCC patients with intermediate stage HCC which cannot be treated surgically [67,68,69]. Examples of chemotherapeutic agents delivered by drug-emitting bead transcatheter arterial chemoembolization include doxorubicin, cisplatin, oxaliplatin and arsenic trioxide. Thus, drug-emitting bead transcatheter arterial chemoembolization offers a drug delivery mechanism which should enhance the ability to target HCC tumors, and hence reduce effects on non-tumor liver tissue and systemic side effects.

Drug-emitting bead transcatheter arterial chemoembolization is a radiological procedure. In principle, this involves the intra-arterial injection to the site of a tumor of a viscous emulsion composed of a drug-emitting bead (e.g., CalliSpheres microspheres) mixed with iodized oil to deliver one or more chemotherapeutic agents to the site of the tumor. This is followed by embolization of the blood vessel with gelatine sponge particles which confines the chemotherapeutic agent to the vicinity of the tumor and creates an ischemic environment which assists in killing HCC cells [67,68,69].

## 6. Cancer Stem Cells and the Initiation and Progression of Hepatocellular Carcinoma

As described above, the liver of a normal adult human harbors liver stem cells (also called progenitor cells) [6,7,8,9,10]. These liver stem cells are identified by a specific set of surface marker proteins. Stem cells/progenitor cells can differentiate to adult hepatocytes and cholangiocytes, which is the normal process for the turnover of these cells (shown schematically in Figure 6). Upon differentiation, liver stem cells/progenitor cells can also provide new hepatocytes for liver regeneration following some forms of liver damage, for example liver surgery. Mitosis of adult hepatocytes also contributes to replacement and expansion of the adult hepatocyte population [7,9,10].

Analysis of cells present in HCC liver tissue has identified a small subset of liver cancer cells called liver cancer stem cells. The properties of these cells are similar to those of the pools of cancer stem cells identified in many other cancers. Liver cancer stem cells are identified experimentally by a specific set of surface marker proteins, for example EpCAM, CD133, CD44, CD90, OV-6, ALDH, K19, which also distinguish them from “normal” liver stem cells/progenitor cells [7,8,9,10]. Liver cancer stem cells are characterized by their abilities to self-renew, differentiate, rapidly proliferate, and their multilineage potential. Liver cancer stem cells are thought to be involved in the initiation of HCC, recurrence of HCC following treatment, HCC metastasis and in the development of the resistance of HCC to therapies. There is now much evidence to suggest that the destruction of liver cancer stem cells is a desirable target in the treatment of HCC [7,8,9,10].

Liver cancer stem cells are thought to arise from liver stem cells/progenitor cells and also from adult hepatocytes (shown schematically in Figure 6) [7,8,9,10]. There is some evidence that liver cancer stem cells are physically located in “niches” in the liver. Under conditions which foster the initiation and promotion of HCC (chronic inflammation, HBV, HCV, alcohol, non-alcoholic fatty liver disease), liver cancer stem cells can undergo further genetic and epigenetic changes leading to HCC cells and HCC or intrahepatic cholangiocarcinoma.

The relationships between liver stem/progenitor cells, adult hepatocytes, cholangiocytes, liver cancer stem cells and HCC cells are complex, difficult to fully deduce from current observations and hence presently only partly understood. Therefore, both the discussion above and the scheme shown in Figure 6, should be viewed as a guide only.

## 7. Methodology and Terminology for Hepatocellular Carcinoma Liver Samples and Hepatocellular Carcinoma Cell Lines

Research directed towards understanding the role of intracellular Ca^2+^-signaling in the development and progression of HCC has historically employed hepatocytes and liver cell lines. These were derived principally from rat and mouse liver tumors. Over the past ten years or so, most studies have employed cell lines derived from human liver and human HCC tissue, mouse chemical or xenograft models of HCC tumors and the analysis of tissue samples taken from the livers of HCC patients. Examples of cell lines derived from human HCC tumors and from normal human liver and mouse models of HCC employed in the studies addressed in this review are listed in Table 2.

The process of obtaining human liver HCC tissue usually involves taking a tissue sample from a liver resection (such as that shown in Figure 5A), then the selection tissue samples taken from the actual tumor and from “non-tumor” (para-tumor) tissue obtained from outside the margins of the tumor (Figure 5B). The ideal situation is where both HCC tumor tissue and non-tumor (para-tumor) tissue is taken from the same liver. In the discussions below, the terms HCC liver tissue and non-tumor tissue will be employed to describe samples taken from the livers of HCC patients, without reference to exactly how the “control” non-tumor tissue was obtained. As implied in the discussion of liver cancer stem cells above, the para-tumor liver tissue may or may not be the same as “normal” liver tissue. Moreover, while hepatocytes or transformed hepatocytes (HCC cells), will be the main cell type present, these tissues will also contain all other cell types present in liver.

A number of HCC cell lines derived from human HCC or “normal” liver are listed in Table 2. These have varying degrees of dedifferentiation from normal adult hepatocytes. Some studies have also employed “normal” liver cell lines such as L01 and L02 liver cell lines [72]. These are derived from adult or fetal human liver. However, they are “immortalized” cells, so caution is needed in treating these as truly “normal” hepatocytes. In addition, constant vigilance is required to be assured that the cell line employed in a given study is true to type and is not contaminated by other cells [73].

## 8. Mutations and Altered Expression of Ca^2+^-Signaling Proteins in Hepatocellular Carcinoma

In considering Ca^2+^ signaling proteins as therapeutic targets in the prevention and treatment of HCC, it is useful to summarize the mutations and altered expression of Ca^2+^ signaling proteins so far detected in HCC in the human liver. Our current knowledge of these mutations and alterations in the expression of Ca^2+^ signaling proteins is summarized in Table 3 and Table 4. The data were obtained from an analysis of DNA and mRNA and/or protein expression present in HCC and “normal” liver tissue obtained from HCC patients. These show that mutations and altered gene expression have been detected in a wide range of Ca^2+^ channels, transporters and Ca^2+^-binding proteins. Moreover, comparison of the genes listed in Table 3 and Table 4 with the Ca^2+^-signaling proteins listed in Table 1 as potential targets for treatment of HCC indicates that many potential Ca^2+^-signaling targets are also mutated or under- or overexpressed in HCC.

As discussed above, the liver is heterogeneous with respect to cell types. Therefore, it is expected that a given HCC liver sample will contain a mixture of normal hepatocytes, HCC cells, liver progenitor/stem cells, liver cancer stem cells, stellate cells, Kupffer cells, sinusoidal endothelial cells, cholangiocytes and likely some other cell types. Thus, the observed mutations could be in the DNA of any one or more of these cell types. Nevertheless, the predominant cell types in HCC tissue removed for analysis are likely to be hepatocytes and transformed hepatocytes (HCC cells). Consequently, it is likely that the observed mutations and alterations in gene expression relate to DNA derived from HCC cells. The application of single-cell transcriptome analysis will help to resolve the liver cell types responsible for altered gene expression in HCC [76].

## 9. Voltage-Operated Ca^2+^ Channels, InsP_3_ Receptors and TRPV2 Channels in Liver Cancer Stem Cells

The results of several recent studies point to the importance of intracellular Ca^2+^-signaling, voltage-operated Ca^2+^ channels and InsP_3_Rs in maintaining the characteristics of liver cancer stem cells and the ability of these cells to self-renew (Table 1). In a search for marker proteins which could be used to identify liver cancer stem cells, Zhao and colleagues generated a monoclonal antibody, IB50-1, which turned out to bind to isoform 5 of α2δ1, a subunit of several voltage-operated Ca^2+^ channels [13]. Analysis of HCC cells isolated from frozen tissue sections obtained from HCC liver resections showed a subgroup of IB50-1 positive cells which were enriched in HCC tissue compared with non-tumor tissue. The IB50-1 antibody thus allowed the authors to identify tumor initiating cells with the properties of liver cancer stem cells. From these cell suspensions they created a stable cell line, Hep-12, of liver cancer stem cells which all expressed the α2δ1 protein and a stable cell line, Hep-11, which did not express the α2δ1. Hep-12 cells exhibited enhanced self-renewal properties, including spheroid formation (an indicator of self-renewal capacity) and initiated tumors in NOD/SCID HCC mice, whereas Hep-11 cells did not (Figure 7A,B).

Studies of Ca^2+^-signaling using fluo-4 showed that basal [Ca^2+^]_cyt_ in Hep-12 cells was higher than that in Hep-11 cells [13]. Moreover, Hep-12 cells exhibited spontaneous oscillations in [Ca^2+^]_cyt_ which were not observed in Hep-11 cells (Figure 7C,D). Pretreatment of Hep-12 cells with antibody IB50-1 reduced basal [Ca^2+^]_cyt_ and reduced the amplitude of oscillations in [Ca^2+^]_cyt_, while overexpression of α2δ1 in Hep-11 cells increased basal [Ca^2+^]_cyt_. Some evidence for the expression of functional L- and N-type voltage-operated Ca^2+^ channels in Hep-12 cells was obtained using specific voltage-operated Ca^2+^-channel blockers and RT–PCR. In Hep-12 cells, inhibition of α2δ1 by antibody IB50-1 or knockdown of α2δ1 using shRNA induced apoptosis by a mechanism involving decreased phosphorylation of ERK1/2, decreased Bcl2, increased Bax and Bad and activation of caspases 3,8 and 9 (pathway scheme shown in Figure 3). It was concluded from these studies that α2δ1 and hence L- and N- type voltage-operated Ca^2+^ channels, expressed in tumor initiating cells (liver cancer stem cells), promote/maintain the self-renewal characteristics of these cells. Targeted inhibition of α2δ1 should induce apoptosis in liver cancer stem cells and hence reduce the development of HCC. This could be especially useful in preventing the recurrence of HCC tumors, following liver surgery or other initial treatment [13].

Further investigations using Hep-12 (subset of HCC cells enriched in liver cancer stem cells) and Hep-11 (HCC cells) provided additional evidence that oscillations in [Ca^2+^]_cyt_ play a role in maintaining the ability of liver cancer stem cells to self-renew and form tumors and identified InsP_3_R2 as a contributor to these [Ca^2+^]_cyt_ oscillations [16]. Agents expected to enhance the self-renewal of liver cancer stem cells, EGF/FGFβ/B27 and Il-6, induced oscillations in [Ca^2+^]_cyt_ in Hep12 cells, but not in Hep-11 cells, suggesting that oscillations in [Ca^2+^]_cyt_ are an important characteristic of liver cancer stem cells. The authors also employed the genetically encoded GCaMP-ER2 Ca^2+^ sensor targeted to the ER to investigate oscillations in [Ca^2+^]_ER_ and the role of the ER and InsP_3_Rs in the generation of [Ca^2+^]_cyt_ oscillations. Using RNA-seq, 10 genes encoding Ca^2+^-signaling proteins which were expressed at higher levels in Hep-12 cells compared with Hep-11 cells were identified [16]. These included voltage-operated Ca^2+^ channel subunits and InsP_3_Rs. Using sh-RNA to knockdown the expression of several of these genes, InsP_3_R2 was identified as important for the generation of [Ca^2+^]_cyt_ oscillations in Hep-12 cells. Knockdown of InsP_3_R2 in Hep-12 cells reduced their capacity to form spheroids and reduced their capacity to form tumors in NOD/SCID mice.

The results of all these studies on liver cancer stem cells, identified by expression of the α2δ1 antigen, provide evidence that voltage-operated Ca^2+^ channels in the plasma membrane and IP_3_R2 in the ER are important in maintaining the stem cell characteristics of these cells. Inhibition of voltage-operated Ca^2+^ channels and/or InsP_3_R2 may provide a strategy to inhibit the pathologic roles of these cells. Further studies are warranted to gain a deeper understanding of the mechanisms involved, especially the roles of plasma membrane Ca^2+^ entry channels. For example, how expression of L- and N- type voltage-operated Ca^2+^ channels relates to the observation that liver cancer stem cells are depolarized relative to normal stem cells [84].

The α2δ1 protein could also be a potential biomarker for HCC. Badr and colleagues used ELISA to measure the α2δ1 peptide in the serum of patients with chronic HCV infection diagnosed with HCC and in the serum of healthy controls [85]. Serum concentrations of α2δ1 were compared with those of alpha-fetoprotein and annexin A2. A good correlation was observed between serum levels of α2δ1 and alpha-fetoprotein (but not with annexin-1, another putative marker protein) and the stage of HCC.

T-type voltage-operated Ca^2+^ channels may also be involved in regulating the proliferation and differentiation of liver cancer stem cells. Radiofrequency electromagnetic fields have been shown to induce tumor shrinkage in several cancers, including HCC (treatment options, Figure 4) [14]. In a study of the mechanisms involved, employing mouse HCC xenograft tumors and Huh-7 liver cells, Jimenez and colleagues found that the application of an amplitude modulated radiofrequency electromagnetic field (optimized for treatment of HCC tumors), increases Ca^2+^ entry to HCC cells via T-type voltage-operated Ca^2+^ channels, resulting in an increase in [Ca^2+^]_cyt_ and cell death.

It has also been suggested that TRPV2 is a possible target for the therapeutic destruction of liver cancer stem cells [15,86]. Measurement of the expression of TRPV2 together with liver cancer stem cell marker proteins in several liver cell lines and in HCC liver samples has provided some evidence that increased expression of TRPV2 in HCC cells reduces their ability to act as liver cancer stem cells [86].

These studies with T-type voltage-operated Ca^2+^ channels and TRPV2 in liver cancer stem cells suggest that Ca^2+^-signaling pathways in these cells are worthy of further study. Such study should expand knowledge of their role in the initiation and progression of HCC and further evaluate the possibility that they may be future therapeutic targets.

## 10. Mitochondrial Ca^2+^ and Store-Operated Ca^2+^ Entry in HBV and HCV Infection

Infection of the liver by HBV (DNA virus) or HCV (RNA virus) alters intracellular Ca^2+^-signaling to create an environment that permits replication of the virus. This environment, especially virus-initiated increase in ROS, also favors the initiation and promotion of HCC [18,19,87]. Ca^2+^-induced production of ROS by mitochondria is an important source of this virally induced increase in ROS. Viral infection was shown to increase the transfer of Ca^2+^ from the ER to the mitochondria through mitochondrial membrane associated junctions (MAMs) involving the InsP_3_R, voltage-dependent anion channel (VDAC) and Ero1a proteins [17,18]. In the case of HCV this Ca^2+^ movement seems to be initiated by HCV-induced ER stress leading to the unfolded protein response [18]. In the case of HBV, Ca^2+^ movement from ER to mitochondria is initiated by the viral protein HBx, which binds to the voltage-dependent anion channel 3 (VDAC3) [18]. The resulting increase in [Ca^2+^]_MT_ decreases the mitochondrial membrane potential, inhibits the electron transport chain and oxidative phosphorylation and increases the production of ROS and reactive nitrogen species by components of the electron transport chain [17,18]. Pharmacological inhibition of mitochondrial Ca^2+^ uptake has been proposed as a strategy to inhibit the downstream consequences of HBV and HCV infection, including the initiation of HCC [17].

HBV-initiated synthesis of the HBV Bax1 protein and the activation of Ca^2+^ uptake to mitochondria by Bax1 have also been implicated in the activation of store-operated Ca^2+^ entry in hepatocytes infected with HBV [19]. Expression of Bax1 increased store-operated Ca^2+^ entry in isolated rat hepatocytes in culture transfected with DNA encoding Bax1. ATP was used to release Ca^2+^ from the ER via production of InsP_3_ and a “Ca^2+^ add back” protocol and Fura-4 F fluorescence Ca^2+^ imaging were employed to measure store-operated Ca^2+^ entry. No change in the expression of Orai, STIM or TRP proteins (components of the store-operated Ca^2+^ entry pathways) was detected by RNA-seq. The authors suggested that Ca^2+^ uptake by mitochondria enhanced by HBV Bax1 reduces Ca^2+^-mediated feedback inhibition of store-operated Ca^2+^ entry thus increasing Ca^2+^ entry [19].

## 11. Store-Operated Ca^2+^ Entry, SERCA2b and Ca^2+^/Calmodulin-Dependent Protein Kinases in Initiation and Progression of Hepatocellular Carcinoma in Non-Alcoholic Fatty Liver Disease

As for hepatitis viral infection, a chronic increase in ROS promotes the initiation and progression of non-alcoholic fatty liver disease to non-alcoholic steatohepatitis and then to HCC [88,89]. Increased production of ROS is caused by an accumulation of excess lipid in cytoplasmic lipid droplets in hepatocytes [88,89]. Several Ca^2+^ channels, transporters and binding proteins are thought to be important in the initiation and progression of HCC (Table 5). These Ca^2+^-signaling proteins and pathways regulate the accumulation and removal of lipid from lipid droplets and lipid-induced production of ROS. They offer potential targets for therapeutic strategies directed towards normalizing intracellular Ca^2+^-signaling in hepatocytes, reducing cytoplasmic lipid and ROS, and hence preventing the initiation and promotion of HCC. The nature and rationale for identifying these targets has been reviewed elsewhere [5].

## 12. STIM1 and Orai1 in the Progression and Metastasis of Hepatocellular Carcinoma

STIM1, Orai1 and store-operated Ca^2+^ entry were shown to be important in HCC proliferation and metastasis and to be involved in mediating the ability of anticancer drugs to kill HCC cells (Table 1). However, the roles played by STIM1 and Orai1 appear complex and are presently only partly understood.

In an early study Yang and colleagues showed that the expression of STIM1 is higher in HCC tumor tissue than in paired non-tumor liver tissue [80]. They measured STIM1 expression in several HCC cell lines and found highest expression in the HCC-LM3 cell line, which also exhibited the highest migration ability. Inhibition of store-operated Ca^2+^ entry with SKF-96365 in HCC-LM3 cells or knockdown of STIM1 with siRNA decreased migration and invasion, assayed in vitro. It was concluded that inhibition of store-operated Ca^2+^ entry is a potential strategy to inhibit HCC migration and invasion.

Several experimental approaches have shown that overexpression of STIM1, leading to enhanced Ca^2+^ entry via store-operated Ca^2+^ channels, promotes the proliferation and metastasis of HCC cells [26]. Knockdown of STIM1 in human SMMC7721 HCC cells, using shRNA, was found to decrease cell proliferation and colony formation, arrest the cell cycle at G0/G1 and decrease DNA synthesis, while overexpression of STIM1 led to enhanced cell proliferation [26].

In studies employing human HepG2 HCC cells and a xenograft nude mouse model of HCC, Li and colleagues showed that hypoxia induced the expression of STIM1, increased [Ca^2+^]_cyt_ and increased store-operated Ca^2+^ entry [95]. These changes were associated with increased expression of transcription factor HIF-1. Increased expression of STIM1 was also observed in HCC tumors in mice and in samples of human HCC tissue compared with non-tumor tissue [95]. Knockdown or overexpression of STIM1 were associated with reduced and increased tumor growth, respectively. The mechanisms involved were further investigated using YC-1, an inhibitor of HIF-1. It was concluded that the growth of HCC tumors leads to a decrease in O_2_ available to HCC cells, activation of HIF-1 and increased transcription of STIM1, mediated by the binding of HIF-1 to the HRE motif in the STIM1 promoter. This, in turn, increases store-operated Ca^2+^ entry and [Ca^2+^]_cyt_ and activates p300 and CamKII, leading to the stabilization of HIF-1. This then re-enforces the increased transcription of STIM1 (shown schematically in Figure 8). While the central role of increased [Ca^2+^]_cyt_ and STIM1 expression seem well established, further experiments may be needed to elucidate: the mechanisms involved in the first step (initial activation of HIF1 by hypoxia); whether factors additional to increased STIM1 protein are required to activate store-operated Ca^2+^ entry; and the downstream mechanisms by which increased STIM1 expression and increased [Ca^2+^]_cyt_ enhance tumor cell growth.

However, as just discussed, overexpression of STIM1 contributes to enhanced HCC cell proliferation and tumor growth, whereas decreased expression of STIM1 may be associated with HCC metastasis [61]. Immunohistochemistry showed decreased expression of STIM1 at the edge of HCC tumors (Figure 5B,C). Using four different human HCC cell lines, grown under conditions which promoted either proliferation or metastasis, Zhao and colleagues found that STIM1 expression was lower in the cells exhibiting metastatic properties [61]. Further studies of the pathways involved led to the conclusion that during the proliferation and growth of HCC cells, STIM1 is expressed at a high level, leading to enhanced store-operated Ca^2+^ entry and activation of the CamKII/Akt/Gsk3β pathway which, in part, maintains expression of transcription factor Snail1 at a low level. In metastatic HCC cells, expression of Snail1 is increased, leading to a decrease in STIM1 expression (mediated by binding of Snail1 to the Ebox of the STIM1 promoter). This is also associated with decreased glycolysis and increased fatty acid oxidation [61].

Studies with TRPC1 have provided indirect evidence for a role for store-operated Ca^2+^ entry in the mechanisms by which cell proliferation is enhanced in HCC cells [96,97]. From the results of studies employing the Huh-7 HCC cell line and shRNA to suppress TRPC1 expression, the authors suggested possible roles for both store-operated Ca^2+^ entry and TRPC1 in the regulation of cell proliferation in HCC cells.

Orai1 may be involved in the progression of HCC and in the mechanisms by which the anticancer drug 5-fluorouracil kills HCC cells [27]. Expression of Orai1 was found to be increased in HCC tissue compared with that in non-tumor liver tissue. Experiments employing HepG2 cells provided evidence that 5-flurouracil induces autophagic cell death by inhibiting the PI3K/Akt/mTOR pathway. 5-Fluorouricil also decreased store-operated Ca^2+^ entry, reduced the expression of Orai1, but did not alter STIM1 or TRPC1 expression. Knockdown of Orai1 or pharmacological inhibition of store-operated Ca^2+^ entry, inhibited the PI3K/Akt/mTOR pathway and potentiated autophagic cell death. Increased expression of Orai1 lessened the ability of 5-fluorouracil to induce autophagic cell death. It was concluded that inhibition of Orai1, coincident with treatment with 5-fluorouracil, may enhance the sensitivity of HCC cells to the drug [27].

All these studies identify store-operated Ca^2+^ entry to HCC cells as an important signal for proliferation and metastasis and a potential target for therapeutic intervention—especially in the reduction of tumor growth. However, further studies are needed in order to gain a more complete understanding of the role of store-operated Ca^2+^ entry at different stages of HCC tumor development.

## 13. TRPC6, TRPV4 and TRPV1 in the Progression, Metastasis and Apoptosis of Hepatocellular Carcinoma

Three TRP non-selective cation channels have so far been identified as potential therapeutic targets for HCC. These are TRPC6 and TRPV4, implicated in mediating progression and metastasis, and TRPV1 which offers an avenue to kill HCC cells via Ca^2+^-signaling (Table 1). Some time ago, El Boustany and colleagues investigated the role of TRPC6-mediated Ca^2+^ entry in HCC cell proliferation in a study which employed Huh-7 HCC cells as well as hepatocytes isolated from liver tissue obtained from HCC patients [28]. Overexpression of TRPC6 was found to enhance cell proliferation and store-operated Ca^2+^ entry (measured using Fura-2 and a “Ca^2+^ add back” protocol) while TRPC6 knockdown inhibited cell proliferation and reduced store-operated Ca^2+^ entry. Knockdown of STIM and Orai also reduced store-operated Ca^2+^ entry. The expression of TRPC6 in hepatocytes isolated from HCC liver tissue was found to be somewhat greater than that in hepatocytes isolated from normal liver tissue. The authors concluded that these results provide evidence that Ca^2+^ entry to HCC cells via TRPC6 plays a role in promoting the proliferation of HCC cells and oncogenesis [28].

In experiments employing HepG2 and Huh-7 HCC cells, Xu and colleagues found that TRPC6 and the plasma membrane (Na^+^-Ca^2+^) exchange protein (NCX1) are required components of the pathway by which the cytokine TGFβ enhances HCC tumorigenesis and progression [98]. TGFβ was found to induce the phosphorylation of Smad proteins, which regulate gene transcription [98]. The authors found that TGFβ induces an increase in [Ca^2+^]_cyt_ in HCC cells. The use of inhibitors of TRPC6 channels and of the (Na^+^-Ca^2+^) exchanger or shRNA to knockdown the expression of these proteins, showed that they are required for TGFβ -induced migration, invasion and metastasis in HCC cells. Co-immunoprecipitation experiments provided evidence that TGFβ increased the association of TRPC6 protein with the (Na^+^-Ca^2+^) exchange protein. In human HCC tissues, the expression of both TRPC6 and the (Na^+^-Ca^2+^) exchanger was higher in human HCC tissue compared to that in non-tumor tissue. Moreover, higher levels of expression of these proteins correlated with a higher tumor grade [98].

TRPC6 has been identified as a target of the 21 amino acid terebrid snail venom peptide (Tv1), which was found to inhibit cell proliferation and migration in BNL1MEA.7R.1 (1MEA) cells, a mouse liver carcinoma cell line and to reduce tumor size in tumor bearing mice [29]. Expression of TRPC6 (and TRPV1) was also found to be higher in mouse 1MEA HCC cells than that in BNL.CL.2 cells, a mouse liver epithelial cell line. Fluorescence imaging in 1MEA cells showed that Tv1 colocalizes with TRPC6. Further studies of the pathways involved, as well as molecular modeling of the interaction of Tv1 with TRPC6, led to the conclusion that in HCC cells, Tv1 binds to TRPC6 leading to decreases in: [Ca^2+^]_cyt_; NFAT activation; cyclooxygenase-2 (COX-2) expression; and PGE_2_ concentration, all of which contribute to inhibition of HCC cell proliferation and migration [29].

Elevated expression of TRPV4 has also been observed in human HCC tissue compared to that in non-tumor liver tissue [30]. Moreover, pharmacological inhibition of TRPV4 with HC-067,047 in a HCC cell line decreased cell proliferation, increased apoptosis and inhibited the epithelial-mesenchymal transition [30]. In a NOD-SCID mouse xenograft HCC model, intraperitoneal injection of HC-067,047 also inhibited tumor growth and induced apoptosis. Inhibition of TRPV4 has also been shown to reduce liver fibrosis, a risk factor for HCC [99]. Using a mouse model of CCl_4_-induced liver fibrosis, Fu and colleagues employed the TRPV4 agonist GSK1016790A and inhibitor HC-067,047 in order to activate and inhibit, respectively, TRPV4. Their results provide evidence that activation of TRPV4 contributes to the development of liver fibrosis. Taken together, the studies above suggest that targeting TRPC6 or possibly TRPV4 with a pharmacological inhibitor could be used to inhibit of HCC progression.

Pharmacological activation of TRPV1 provides a potential avenue for killing HCC cells, as suggested for some other cancers [31,32,100]. In many cell types, the activation of TRPV1 by capsaicin or other agonists enhances a large Ca^2+^ entry which can induce apoptosis [31,32,100]. Capsaicin offers the advantage of being a natural product and like other potential agents, could be physically targeted to a liver tumor using drug-emitting bead transcatheter arterial chemoembolization (discussed above). Numerous studies, principally all employing HepG2 cells, have shown that capsaicin induces apoptosis in HCC cells [31,101,102]. The mechanisms involved include an increase in [Ca^2+^]_cyt_ and ROS and activation of the STAT3 pathway [31,101,102]. Capsaicin has also been shown to act synergistically with the multikinase inhibitor sorafenib and with the application of a static magnetic field, in reducing the proliferation of HCC cells [31,100]. Capsaicin enhanced HCC cell death induced by sorafenib or magnetic field and in the case of the application of capsaicin in combination with a magnetic field, cell death was reduced by the TRPV1 inhibitor SB-705,498 [31].

Most of the studies above focused on the ability of capsaicin to kill HCC cells, without necessarily linking the actions of capsaicin to the expression of TRPV1 in these cells. In an early study, Miao and colleagues reported that the TRPV1R protein is expressed in human HCC tissue and provided some evidence that higher expression of TRPV1R is associated with better HCC prognosis [103]. More recently, the role of TRPV1 in HCC has been studied using TRPV1 KO mice, diethylnitrosamine and xenograft mouse models of HCC and HCC cell lines [32]. TRPV1 was identified in HepG2 and Huh-7 HCC cells and in human liver HCC tissue. Knockout of TRPV1 promoted tumorigenesis and noticeably altered liver histology, while administration of capsaicin to mice inhibited tumor growth. The authors suggested that TRPV1 is a potential therapeutic target for HCC.

From these studies it can be concluded that Ca^2+^ entry through TRPC6 and TRPV4 contributes to the growth and progression of HCC, while activation of TRPV1 may be targeted to kill HCC cells. However, substantial further research is needed to clarify the roles of each of these channels, especially the potential of TRPV1 to mediate HCC apoptosis.

## 14. Type 3 InsP_3_ Receptors in the Progression of Hepatocellular Carcinoma

Two recent studies have shown that Ca^2+^ release from the ER through InsP_3_R3 mediates the progression of both HCC and cholangiocarcinoma [33,46]. In hepatocytes isolated from normal human liver, expression of InsP_3_R3 was found to be very low or not detectable. Immunohistochemistry showed increased expression of InsP_3_R3 in human HCC tissue compared with non-tumor liver tissue and increased expression in HCC cell lines [33]. High expression in HCC tissue was associated with poorer patient outcomes. CRISPR/Cas9 was used to delete InsP_3_R3 from HepG2 cells, and these were then used to grow tumors in a nude mouse xenograft model. Tumors derived from InsP_3_R3 KO HepG2 cells grew less than those derived from WT HepG2 cells. The InsP_3_R3 gene was found to be strongly methylated in normal human liver tissue but was demethylated at many sites in HCC tissue. Mice treated with a demethylating agent exhibited increased expression of InsP_3_R3 in specific regions of the liver and this was associated with increased InsP_3_-mediated Ca^2+^-signaling and enhanced cell proliferation and liver regeneration following hepatectomy [33].

Expression of InsP_3_R3 was also found to be elevated in both hilar and intrahepatic samples from human cholangiocarcinoma liver and in cholangiocarcinoma cell lines compared to normal cholangiocytes [46]. Deletion of InsP_3_R3 from cholangiocarcinoma cell lines decreased proliferation and migration. In MzChAl cholangiocarcinoma cells, InsP_3_R3 was found to be localized in regions of the ER close to the mitochondria, as well as in apical regions, where it is principally located in normal cholangiocytes. Increased expression of InsP_3_R3 induced an increase in [Ca^2+^]_MT_ while deletion of InsP_3_R3 impaired the increase in [Ca^2+^]_MT_ and caused cell death. The results of these studies with both HCC and cholangiocarcinoma suggest that targeting InsP_3_R3 with a pharmacological inhibitor would inhibit progression of HCC [33,46].

## 15. The Mitochondrial Ca^2+^ Uniporter, Permeability Transition Pore and Mitofusin-2 in Hepatocellular Carcinoma Metastasis

In liver cells, as in most other cell types, [Ca^2+^]_MT_ plays important roles in the regulation of the citric acid cycle and ATP production and in the apoptotic pathway. Under normal conditions, Ca^2+^ is transported from the cytoplasmic space into the mitochondrial matrix via the mitochondrial Ca^2+^ uniporter. Ca^2+^ can also be transferred directly from the ER to the mitochondrial matrix via the mitochondrial-associated membranes (MAMs) and the voltage-dependent anion channel (VDAC). Ca^2+^ is normally transported out of mitochondria by the (Ca^2+^ - H^+^) exchanger [104]. When [Ca^2+^]_cyt_ and/or [Ca^2+^]_ER_ is chronically elevated the mitochondrial matrix may become overloaded with Ca^2+^. This then triggers the release of Ca^2+^ via the mitochondrial permeability transition pore (pathway shown schematically in Figure 3B) [34,81,104]. Uptake of Ca^2+^ through the mitochondrial Ca^2+^ uniporter is regulated by a number of proteins, including mitochondrial Ca^2+^ uniporter regulator 1 (MCUR1) and mitochondrial Ca^2+^ uptake 1 and 2 (MICU1 and 2) [34,81,104]. Several studies have shown that both the mitochondrial uniporter and mitochondrial permeability transition pore play important roles in HCC progression and metastasis [34,35,36,81,105].

In human HCC tissue, the mitochondrial Ca^2+^ uniporter protein was found to be expressed at higher levels than in paired non-tumor tissue. Higher expression correlated with a higher risk of recurrence and death in HCC [34]. Studies conducted using several HCC cell lines in which expression of the mitochondrial Ca^2+^ uniporter was experimentally either reduced or increased showed that high levels of mitochondrial Ca^2+^ uniporter enhance the uptake of Ca^2+^ to mitochondria and increase [Ca^2+^]_MT_, which in turn increases ROS and promotes HCC metastasis. Buffering of [Ca^2+^]_MT_, using parvalbumin genetically targeted to the mitochondrial matrix (PV-Mito), suppressed the production of ROS and development of metastatic properties. Further analysis of the pathways involved led to the conclusion that increased expression of mitochondrial Ca^2+^ uniporter and enhanced uptake of Ca^2+^ to the mitochondrial matrix cause an increase in ROS via changes in NAD/NADH, SIRT3 and superoxidase dismutase 2 (SOD2), as shown schematically in Figure 9 [34].

Two studies provide evidence that increased expression of mitochondrial Ca^2+^ uniporter regulator 1 enhances HCC cell survival via the p53 pathway and also enhances metastasis via the Nrf2 pathway. Expression of mitochondrial Ca^2+^ uniporter regulator 1, was found to be high in many HCC liver samples compared with that on non-tumor liver tissue [81]. Using selected HCC cell lines in which mitochondrial Ca^2+^ uniporter regulator 1 was expressed at either high or low levels, Ren and colleagues found that elevated expression of mitochondrial Ca^2+^ uniporter regulator 1 is associated with elevated [Ca^2+^]_MT_, increased mitochondrial ROS, decreased apoptosis and increased cell proliferation [81]. In a mouse xenograft HCC model, tumors arising from the implantation of HCC cells expressing high levels of mitochondrial Ca^2+^ uniporter regulator 1 exhibited enhanced growth compared with that of tumors derived for HCC cells exhibiting low mitochondrial Ca^2+^ uniporter regulator 1 expression [81]. Further investigation of the pathways involved provided evidence that increased expression of mitochondrial Ca^2+^ uniporter regulator 1 enhanced the proliferation of HCC cells by increasing [Ca^2+^]_MT_ and generation of mitochondrial ROS, leading to enhanced degradation of p53, decreased autophagy and increased cell cycle activity and cell proliferation (shown schematically Figure 9) [81].

Higher levels of mitochondrial Ca^2+^ uniporter regulator 1 have also been shown to promote the epithelial mesenchymal transition in HCC cells [105]. Studies employing HCC cell lines showed that knockdown of mitochondrial Ca^2+^ uniporter regulator 1 and the associated decreased mitochondrial and cellular production of ROS, led to decreased invasion and migration, increased expression of epithelial cell markers (ZO-1 and E- cadherin), decreased expression of mesenchymal markers (N- cadherin and vimentin) and decreased expression of the transcription factor snail, a regulator of the epithelial mesenchymal transition [105]. Overexpression of mitochondrial Ca^2+^ uniporter regulator 1 caused the opposite changes. Measurement of the expression of mitochondrial Ca^2+^ uniporter regulator 1 in human HCC tissue samples indicated that higher levels of mitochondrial Ca^2+^ uniporter regulator 1 were associated with more advanced HCC. From these studies it was concluded that increased expression of mitochondrial Ca^2+^ uniporter regulator 1 increases [Ca^2+^]_MT_, ROS production, activates Nrf2 and the Notch pathway which mediates the epithelial mesenchymal transition (Figure 9) [105]. Taken together, these studies suggest that targeting the mitochondrial uniporter protein or one of its regulator proteins with a pharmacological inhibitor should inhibit the progression of HCC.

Enhanced opening of the mitochondrial permeability transition pore leading to increased apoptosis has been implicated in the mechanism by which the natural product erinacine, obtained from the mushroom *Hericium erinaceus*, inhibits the growth of HCC cells [35]. Expression of components of the PI3kinase/Akt/GSK-3β pathway was found to be increased in human HCC tissues compared to that in non-tumor liver tissue. Experiments with HepG2 HCC cells using the PI3kinase inhibitor LY294002 showed that inhibition of PI3kinase decreased the mitochondrial membrane potential and increased [Ca^2+^]_MT_ (measured using Rhodamine 2). In HepG2 cells, erinacine decreased cell proliferation, colony formation, migration and invasion. These changes were associated with decreases in mitochondrial membrane potential, expression of components of the PI3kinase/Akt/GSK-3β pathway, cyclin D, vimentin, β-catenin and Bcl2 and increased expression of E- cadherin. Further, erinacine increased apoptosis, as indicted by increased Bax and caspase-9. Erinacine also enhanced the action of LY294002 on these pathways. In a mouse xenograft liver tumor model, erinacine reduced tumor size [35]. These results provide some insight into the ability of erinacine to inhibit the growth and progression of HCC and implicate [Ca^2+^]_cyt_, [Ca^2+^]_MT_, the mitochondrial permeability pore transition and the components of the PI3kinase/Akt/GSK-3β pathway in the mechanism of erinacine action. However, further experiments are needed to define the exact roles of [Ca^2+^]_cyt_, [Ca^2+^]_MT_ and the mitochondrial permeability pore transition.

Activation of mitofusin-2 has been identified as a target for reduction of HCC through enhancing the transfer of Ca^2+^ from the ER to mitochondria, thereby inducing apoptosis [36]. Mitofusin-2 is located in the mitochondrial outer membrane and is involved in mitochondrial fusion [36]. Increased expression of mitofusion-2 in HCC cells induces apoptosis [36]. Wang and colleagues found that expression of mitofusin-2 in HCC tissue is lower than that in non-tumor liver tissue. Lower mitofusin-2 expression was found to be associated with poorer outcomes. Overexpression of mitofusin-2 in HepG2 HCC cells induced apoptosis and this was blocked by the combination of Ru360, an inhibitor of mitochondrial Ca^2+^ uptake and heparin, an inhibitor of InsP_3_-induced ER Ca^2+^ release. Mitofusin-2 overexpression also reduced the mitochondrial membrane potential, reduced [Ca^2+^]_ER_ and increased [Ca^2+^]_MT_ and cellular ROS. Decreased expression of mitochondrial Ca^2+^ uptake proteins 1 and 2 was also observed in HepG2 cells overexpressing mitofusin-2. The authors concluded that increasing the expression of mitofusin-2 in HCC cells induces apoptosis by enhancing Ca^2+^ entry to mitochondria from the ER [36]. Thus, these studies suggest that targeting mitofusin-2 with a pharmacological activator would inhibit progression of HCC.

## 16. Tuftelin1, Ca^2+^ Calmodulin Kinases and Ca^2+^ Binding Protein 39 in the Promotion and Metastasis of Hepatocellular Carcinoma

The tuftelin1 (TUFT1) protein is involved in the development and mineralization of tooth tissue and its expression is increased in hypoxia [106]. Noting that the tumor microenvironment is often hypoxic, as discussed above, Dou and colleagues investigated the expression and role of tuftelin1 in the growth and metastasis of HCC tumors [106]. Expression of tuftelin1 was found to be increased in HCC liver tissue compared with that in non-tumor tissue. Moreover, increased expression of tuftelin1 correlated with poor overall survival. The pathways involving tuftelin1 in HCC cells were investigated using five different HCC cell lines. Knockdown of tuftelin1 blocked the ability of hypoxia to promote HCC progression, as assessed in HCC cells by monitoring cell growth, metastatic properties and the epithelium mesenchymal transition. Ectopic (increased) expression of tuftelin1 increased [Ca^2+^]_cyt_ (assessed as an increase in the percentage of cells positive for elevated [Ca^2+^]_cyt_) and increased the phosphorylation of Akt, but not that of ERK or JNK. Other experiments identified miR-671-5p as an important regulator of tuftelin1 expression. On the basis of these and other observations, Dou and colleagues proposed the following pathway for the involvement of tuftelin1 and Ca^2+^ in hypoxia-induced HCC progression. Hypoxia, via miR671-5p, activates transcription factor HIF-1α which induces increased expression of tuftelin1. This in turn increases [Ca^2+^]_cyt_, activates the Ca^2+^/PI3kinase/Akt pathway leading to an enhancement of the growth, metastasis and epithelial mesenchymal transition of HCC cells [106].

Experiments directed towards identifying the mechanism by which another micro RNA, miR-129-5p, inhibits HCC growth have identified Ca^2+^/calmodulin-dependent protein kinase IV (CamKIV) as a potential therapeutic target [38]. Expression of miR-129-5p in HCC liver tissue was found to be lower than that in paired non-tumor tissue. In two HCC cell lines, overexpression of miR-129-5p was associated with inhibition of cell proliferation, migration and invasion and the promotion of apoptosis. Using bioinformatics and luciferase reporter assays, the relevant target gene for miR-129-5p was identified as CaMKIV. This protein kinase was found to be expressed at a low level in HCC tissue compared to adjacent non-tumor tissue and was also expressed at a low level in two HCC cell lines, HepG2 and BEL-740 L. Ectopic (increased) expression of CaMKIV in these cell lines inhibited cell proliferation, migration and invasion and promoted apoptosis. In a mouse xenograft model of HCC, in which liver tumors were initiated using HepG2 or BEL-740 L cells, increased expression of CaMKIV inhibited tumor growth. The authors suggest that strategies directed towards increasing the activity of CaMKIV would inhibit the growth of HCC tumors, possibly through a mechanism involving inhibition of the MAPK pathway [38].

About ten years ago, tetrandrine, was shown to inhibit the development of HCC in studies employing HCC cell lines and a mouse HCC tumor model [107]. Tetrandrine is a member of the bisbenzylisoquinoline alkaloid family and is found in the roots of the *Stephania tetrandra* plant. The mechanism by which tetrandrine inhibits HCC involves an increase in ROS and activation of the Akt pathway. In a subsequent study, Huang and colleagues, using structural modeling, identified inhibition of CaMKIIγ as the possible target of tetrandrine action. This was confirmed by measuring the IC_50_ for the inhibition by tetrandrine of cell proliferation and the level of expression CaMKIIγ expression in each of a panel of eight different HCC cell lines [37]. Huh7 cells were most sensitive to tetrandrine and exhibited highest expression of CaMKIIγ, while SMMC-7721 cells were the least sensitive and exhibited the lowest expression of CaMKIIγ. Tetrandrine reduced CaMKIIγ phosphorylation in HCC cells and knockdown of CaMKIIγ reduced the sensitivity of HCC cells to tetrandrine. Expression of CaMKIIγ in human HCC tissue was substantially greater than that in non-tumor tissue [37]. To try to improve the effectiveness of the drug, Lan and colleagues have designed and synthesized a number of derivatives of tetrandrine [108]. These were tested for their ability to induce apoptosis in two HCC cell lines. Compound 31, in which carbon 14 of tetrandrine is substituted with an amide, was the most potent, having an IC_50_ of about 1 µM, which is about 30 times lower than the IC_50_ for the multikinase inhibitor sorafenib [108].

Inhibition of CaMKIIγ may also be a target of berbamine, another member of the bisbenzylisoquinoline family, with a chemical structure similar to that of tetrandrine [15]. Berbamine is isolated from the traditional Chinese herbal medicine plant *Berberis amurensis* [15]. Studies conducted using Huh-7 HCC cells and a NOD/SCID mouse HCC model showed that berbamine inhibits HCC cell proliferation and induces HCC cell death. Knockdown of CaMKIIγ using shRNA or overexpression of CaMKIIγ, provided evidence that a target of berbamine is CaMKIIγ. In human HCC tissue, CaMKIIγ was found to be hyper-phosphorylated compared to the degree of phosphorylation in non-tumor tissue [15]. However, these observations differ from those of Huang and colleagues who compared the effects of berbamine with those of tetrandrine on HCC cells, but did not observe berbamine-induced phosphorylation of CaMKIIγ [37]. From studies conducted using a subpopulation of Huh7 cells selected on the basis of the stem cell marker CD133 (AC133, promimin-1) in which berbamine and its analog 2-methylbenzoyl berbamine, were observed to reduce spheroid formation, Meng and colleagues have also suggested that CaMKIIγ contributes to the maintenance of cancer stem cell characteristics [15].

In further investigation of the role of CaMKIIγ in HCC, Meng and colleagues used bioinformatics to identify protein kinases that regulate gene expression in human HCCs and in a diethylnitrosamine mouse HCC model [109]. One of the kinases identified was CaMKIIγ. Surprisingly, in CaMKIIγ KO mice (CaMKIIγ-/-), the ability of diethylnitrosamine to induce tumors was greatly enhanced compared with that in WT mice. Investigation of the mechanism by which activation of CaMKIIγ inhibits HCC progression provided evidence that, in HCC cells, CaMKIIγ inhibits mTORC and thus inhibits the activation by growth factors of HCC initiation and proliferation via the mTORC pathway [109]. While it seems clear that CaMKIIγ plays important roles in HCC progression, further experiments are required to gain a more complete understanding of these roles and to bring together the observations which suggest that, on one hand, tetrandrine and berbamine inhibit HCC progression by inhibiting CaMKIIγ, while on the other hand, activation of CaMKIIγ actually inhibits HCC progression.

Calcium-binding protein 39 has been shown to be important in the migration and metastasis of HCC cells, suggesting that it may be a potential pharmacological target for the treatment of late stage HCC [39]. Transcriptome sequencing analysis of mRNA from HCC tissue identified calcium-binding protein 39 as potentially important in the progression of HCC. Calcium-binding protein 39 is known to activate several sterol-20 kinases [39]. Expression of calcium-binding protein 39 in human HCC tissue was found to be elevated compared to that in non-tumor liver tissue and higher levels of expression correlated with tumor metastasis and poor survival. Ectopic expression of calcium-binding protein 39 in the immortalized “normal” L02 liver cell line and in several HCC cell lines, increased foci formation and colony formation in soft agar and increased cell mobility. Similar observations were made in a nude mouse model of HCC in which tumors were derived from HCC cells with low and high calcium-binding protein 39 expression. These properties were abolished when expression of calcium-binding protein 39 was knocked down using shRNA. Further studies showed that elevated expression of calcium-binding protein 39 activates the ERK/cJun/Slug signaling pathway to promote the epithelial mesenchymal transition (monitored by measuring increases in N- cadherin and fibronectin and decreases in E-cadherin and α-E-catenin). It was concluded that increased expression of calcium-binding protein 39 in HCC cells enhances the migration and metastatic properties of these cells and hence HCC metastasis [39].

## 17. Conclusions

Numerous Ca^2+^-signaling proteins that could be potential targets for therapeutic treatment of HCC were identified. Targets in liver cancer stem cells, which are thought to drive the initiation and progression of HCC and development of resistance to current drugs, are of particular potential value, as are those which could inhibit the proliferation, migration and metastasis in advanced HCC. However, none of these Ca^2+^-signaling targets has been seriously studied any further than laboratory research experiments. A few of the Ca^2+^-signaling proteins have been identified as targets for natural products previously known to reduce HCC. Promising Ca^2+^-signaling targets seem to be voltage-operated Ca^2+^ channels in liver cancer stem cells, InsP_3_Rs, store-operated Ca^2+^ entry, SERCA2b and Ca^2+^/calmodulin protein kinases. However, for several Ca^2+^-signaling proteins so far identified as potential drug targets, the situation seems complicated, in that the effects of inhibition or activation may depend on the stage of the HCC. The targets so far identified have, to a large extent, evolved from somewhat serendipitous research investigations and from studies of the mechanisms of the actions of natural products. The application of more systematic studies in the future is expected to reveal new Ca^2+^-signaling protein targets and to consolidate priorities for those already identified. Such studies may include expanded genomics and gene expression, RNA-seq and improved knowledge of the fundamental biology and pathology of HCC. If one or more pharmacological inhibitors or activators of Ca^2+^-signaling proteins is shown to be useful in clinical practice, it is likely that such a compound will be most effective in combination with other drugs and will likely be delivered via drug-emitting bead transcatheter arterial chemoembolization.

## Figures and Tables

**Figure 1 cancers-12-02755-f001:**
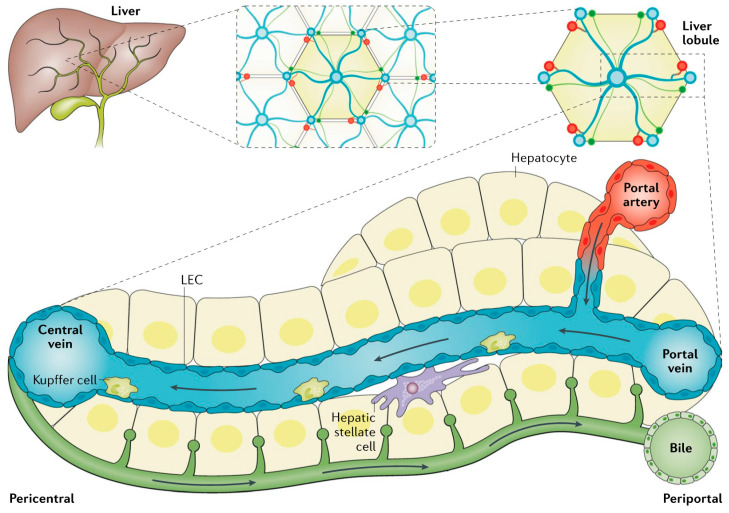
Schematic depiction of the arrangement of hepatocytes and other cell types in a liver lobule and the context of a liver lobule within the whole liver. Blood in the portal artery and portal vein is shown in red and blue, respectively and bile fluid and the bile collecting duct in green. The arrangements of hepatocytes, hepatic stellate cells and liver endothelial cells (LEC) is shown schematically. Taken from Ben-Moshe et al. 2013 [4], with permission.

**Figure 2 cancers-12-02755-f002:**
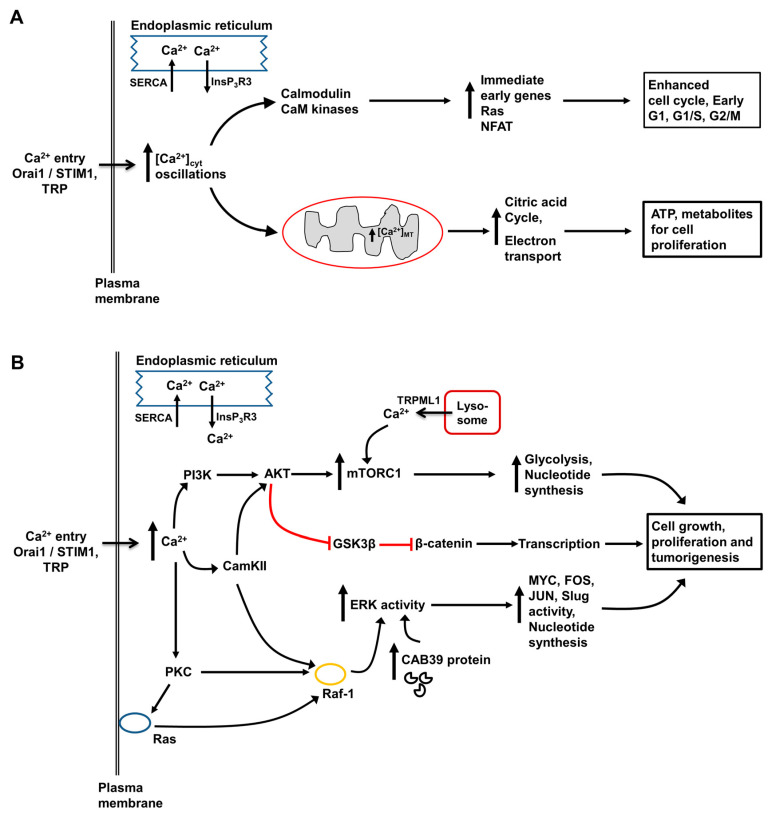
Ca^2+^-signaling pathways and hepatocellular carcinoma (HCC) cell proliferation; (**A**) Schematic representation of pathways involved in Ca^2+^-signaling in HCC cells leading to enhanced cell proliferation; (**B**) schematic representation of the roles of increased [Ca^2+^]_cyt_ in activation of the PI3kinase/Akt/mTORC and ERK pathways in mediating enhanced cell proliferation.

**Figure 3 cancers-12-02755-f003:**
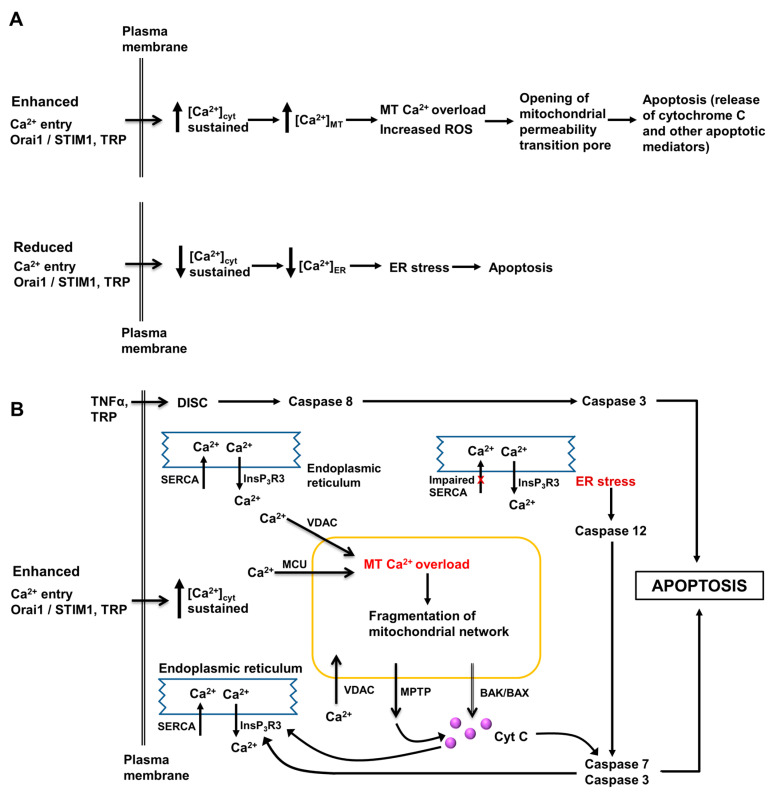
Ca^2+^-signaling pathways and HCC cell apoptosis. (**A**) Schematic representation of pathways involved in Ca^2+^-signaling in HCC cells leading to enhanced cell apoptosis. On one hand, a sustained increase in [Ca^2+^]_cyt_ (e.g., from activation of Ca^2+^ entry) ultimately induces opening of the mitochondrial permeability pore and activation of the apoptotic pathway, while on the other hand sustained decrease in [Ca^2+^]_cyt_ (e.g., from inhibition of Ca^2+^ entry) can lead to ER stress-induced apoptosis; (**B**) scheme showing in more detail the roles of [Ca^2+^]_cyt_, [Ca^2+^]_MT_, mitochondrial and ER Ca^2+^ transporters and channels and caspase enzymes in the apoptotic pathway activated by a sustained increase in [Ca^2+^]_cyt_. VDAC—voltage-dependent anion channel; MPTP—mitochondrial permeability transition pore.

**Figure 4 cancers-12-02755-f004:**
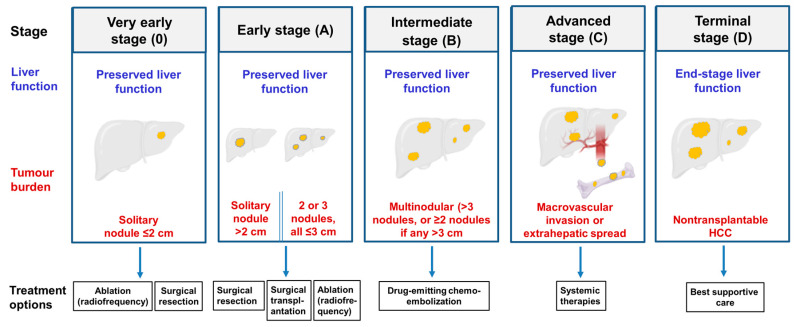
Schematic representation of the stages in the development of HCC. The scheme shows tumor burden, liver function and treatment options for each stage. Adapted from Villanueva 2019 [3].

**Figure 5 cancers-12-02755-f005:**
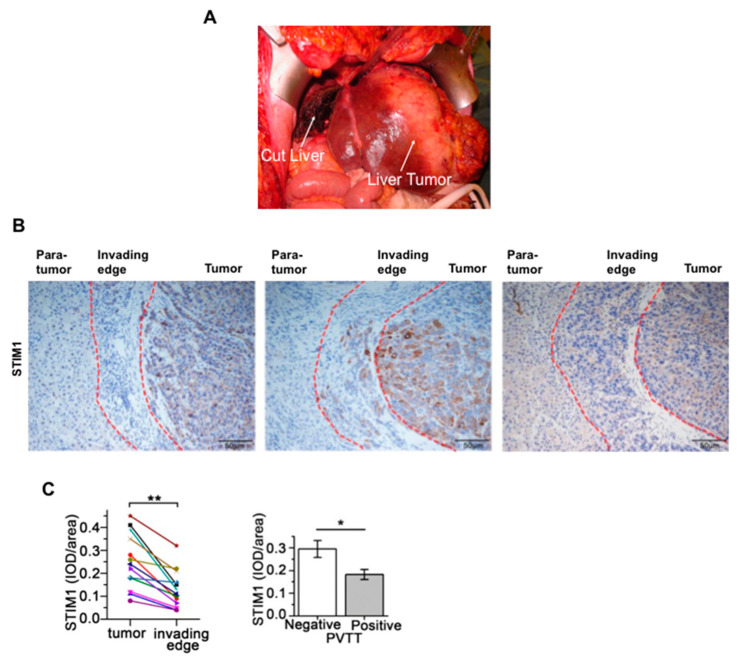
Gross anatomy and histology of HCC liver tumors. (**A**) Removal of an HCC tumor by liver resection. (Photo courteously provided by Dr. R. Padbury, S.A. Liver Transplant Unit, Flinders Medical Center, South Australia.); (**B**) representative micrographs (three separate images) of immunohistochemical analysis (400×) of the expression of STIM1 (identified by red labelling) in HCC tumor tissue and in adjacent non-tumor tissue. Cells enriched in STIM1 protein can be seen both in the tumor and in the invading edge of the tumor; (**C**) integrated optical density (IOD) of STIM1 against immunoglobulin G (IgG) in the tumor and in the invading edge. PVTT, portal vein tumor thrombus. Images in B and graph in C taken from Zhao et al. [61], with permission under Creative Commons License Deed CC BV 4.0), * *p* < 0.05 and ** *p* < 0.01.

**Figure 6 cancers-12-02755-f006:**
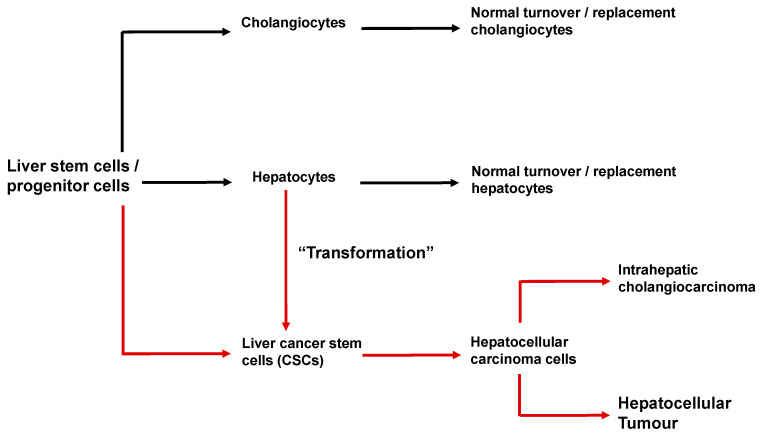
Schematic representation of current ideas on the origin, fate and roles in normal liver and in development of hepatocellular carcinoma and intrahepatic cholangiocarcinoma of liver stem cells/progenitor cells and liver cancer stem cells [7,8,9,10].

**Figure 7 cancers-12-02755-f007:**
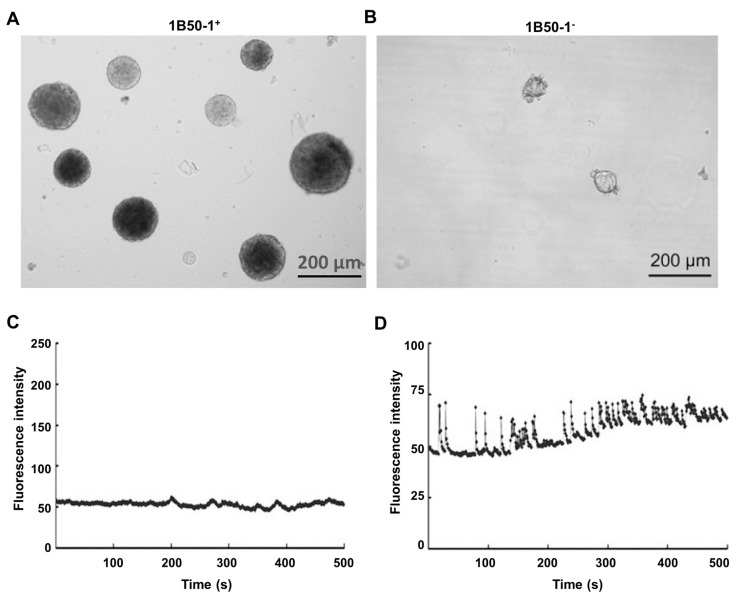
Examples of a self-renewal property and “spontaneous” [Ca^2+^]_cyt_ oscillations in liver cancer stem cells identified as such by expression of the α2δ1 protein (subunit of voltage-operated Ca^2+^ channels). (**A**,**B**) Representative phase-contrast micrographs showing spheroids formed by 1B50-1 positive cells, but not by 1B50-1 negative cells. The α2δ1 protein (antigen) is detected by the 1B50-1 antibody in a subset of liver stem cells (1B50-1 positive). (**C**,**D**) representative plots of [Ca^2+^]_cyt_ (fluorescence of Ca^2+^ sensor) as a function of time for Hep-11 HCC cells (1B50-1 negative) and Hep-12 cells (IB50-1 positive), showing spontaneous oscillations in Ca^2+^ in IB50-1 positive cells (expressing the α2δ1 protein and thought to be liver cancer stem cells). Taken from Zhao et al. 2013 [13], with permission.

**Figure 8 cancers-12-02755-f008:**
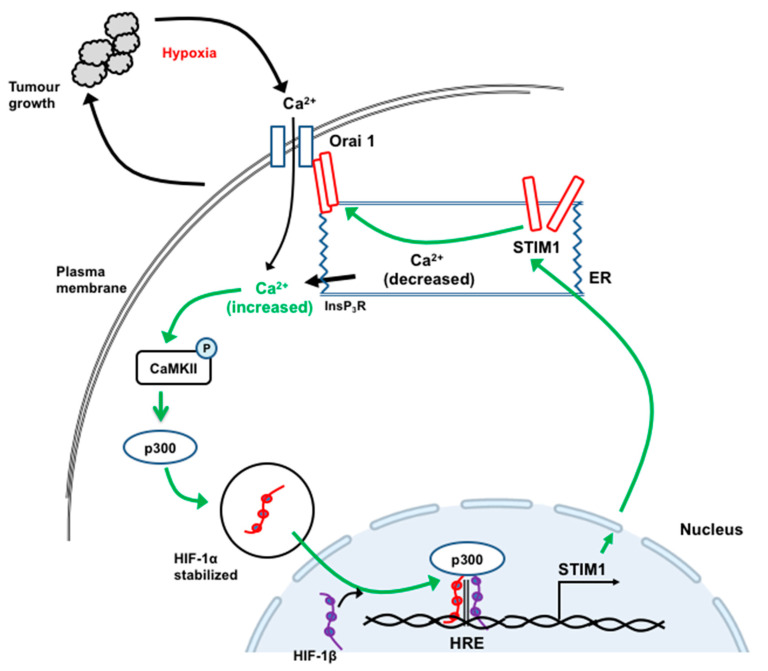
Schematic representation of the proposed mechanism by which the hypoxic environment of HCC tumors is thought to lead to the activation of store-operated Ca^2+^ entry and subsequent increase in expression of STIM1, exacerbation of store-operated Ca^2+^ entry and enhanced tumor growth. It is proposed that hypoxia causes an initial increase in store-operated Ca^2+^ entry which then leads to an increase in [Ca^2+^]_cyt_, activation of CamKII, stabilization of transcription factor hypoxia-induced factor-1 (HIF1), binding of HIF-1 to p300 and the HRE response element of the STIM1 promoter, increased expression of STIM1 and enhanced activation of store-operated Ca^2+^ entry (green arrows). Adapted from Li et al. 2015 [95].

**Figure 9 cancers-12-02755-f009:**
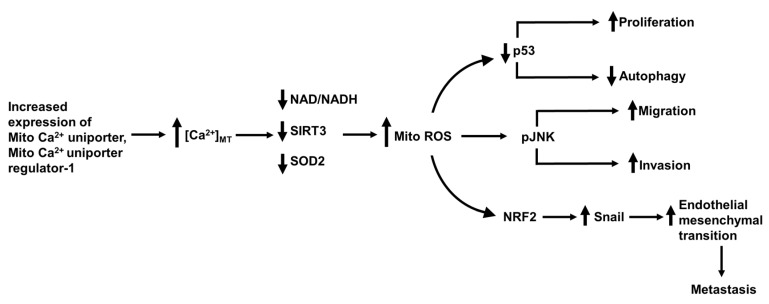
Scheme showing the Ca^2+^-signaling pathways and proteins proposed to be involved in the mechanism by which increased expression and activity of the mitochondrial Ca^2+^ uniporter and mitochondrial uniporter regulator protein 1 lead to increased HCC cell proliferation, decreased apoptosis and increased potential for migration, invasion and metastasis [34,81,105].

**Table 1 cancers-12-02755-t001:** Ca^2+^-signaling proteins so far identified as potential targets for the prevention or treatment of hepatocellular carcinoma.

Stage of Hepatocellular Carcinoma	Targeted Ca^2+^ Signaling Process	Specific Protein Targeted	Proposed Pharmacological Intervention	References
Liver cancer stem cells: initiation, promotion and resistance to systemic drugs	Ca^2+^ entry	L- and N-type voltage operated Ca^2+^ channels	Inhibition	[13]
		T-type voltage operated Ca^2+^ channels	Activation	[14]
		TRPV2 channels	Activation	[15]
	Ca^2+^ release from the endoplasmic reticulum	InsP_3_R2	Inhibition	[16]
	Ca^2+^-binding proteins	CaMKIIγ	Activation	[15]
Hepatitis B and C infection: initiation and promotion	Ca^2+^ uptake by mitochondria	Ca^2+^ transfer from endoplasmic reticulum to mitochondria (voltage-dependent anion channel, VDAC)	Inhibition	[17,18]
	Ca^2+^ entry	Store-operated Ca^2+^ entry	Inhibition	[19]
Non-alcoholic fatty liver disease: initiation and promotion	Ca^2+^ entry	Orai1	Activation	[5,20]
		STIM1		
	Ca^2+^ entry	Voltage-operated Ca^2+^ channels	Inhibition (verapamil)	[21]
	Ca^2+^ uptake by the endoplasmic reticulum	SERCA2b	Activation	[22,23]
	Ca^2+^-binding proteins	CaMKII	Inhibition	[24]
		CaMKK2	Inhibition	[25]
Progression, migration and metastasis	Ca^2+^ entry	Orai1	Inhibition	[26,27]
		STIM1		
		TRPC6	Inhibition	[28,29]
		TRPV4	Inhibition	[30]
		TRPV1	Activation	[31,32]
	Ca^2+^ release from the endoplasmic reticulum	InsP_3_R3	Inhibition	[33]
	Uptake and release of Ca^2+^ from mitochondria	Mitochondrial uniporter (MCU) and MCU regulator protein 1 (Ca^2+^ uptake to mitochondria)	Inhibition	[34]
		Mitochondrial permeability transition pore (Ca^2+^ release from mitochondria)	Activation	[35]
		Mitofusin-2	Activation	[36]
Progression, migration and metastasis	Ca^2+^-binding proteins	CaMKIIγ	Inhibition (tetrandrine, berbamine)	[15,37]
		CaMKIV	Activation	[38]
		Ca^2+^-binding protein 39	Inhibition	[39]
		Increased [Ca^2+^]_cyt_ (tuftelin1)	Inhibition	[39]

**Table 2 cancers-12-02755-t002:** Some models of HCC employed in studies of Ca^2+^-signaling targets in HCC. Liver cells in culture, liver cell lines derived from human hepatocellular carcinoma tumors and mouse liver tumor models.

Model	Attributes of the Model ^a^
Isolated hepatocytes either freshly isolated or in culture	Hepatocytes isolated from human non-diseased liver tissue, subsequently grown in culture for periods of about 1 h to 5 days.
Immortalized human liver cell lines derived from non-diseased human liver	L01 and L02 (HL-7702) liver cells. Immortalized cells originally obtained from normal fetal or adult human liver.
Human HCC cells lines derived from human HCC tissue	Commonly used: HepG2 cells and Huh-7 cells
	Examples of other HCC cell lines include: MHCC97H, SK-Hep-1, SNU398, PLC/PRF/5, SMMC-7721.
Liver cancer stem cells	Hep-12 cells which exhibit liver cancer stem cell marker proteins and Hep-11 control cells which do not exhibit liver cancer stem cell marker proteins;
	Subsets of HCC cells (often HepG2 cells and Huh-7 cells) which exhibit liver cancer stem cell marker proteins.
Mouse liver HCC model in which HCC tumors are induced by a chemical mutagen	Diethylnitrosamine (DEN)-induced liver tumors
Subcutaneous mouse xenograft models	HCC cells (immortalized cell line or cells isolated from human liver HCC tissue) implanted subcutaneously into immunodeficient mice: Nude mice, severe combined immunodeficient (SCID) mice, and non-obese diabetic-severe combined immunodeficiency disease (NOD/SCID) mice

^a^ Summaries of some human HCC cell lines and mouse models of HCC are provided by He et al. 2015 and Caruso et al. 2019 [70,71]. References to some specific human HCC cell lines are also given in the text.

**Table 3 cancers-12-02755-t003:** Mutations detected in Ca^2+^-signaling genes in human HCC tissue.

Ca^2+^-Signaling Pathway	Gene	Protein	References
Ca^2+^ entry channels in plasma membrane	CACNA1B	Voltage-dependent N-type Ca^2+^ channel subunit α-1B	[12]
	CACNA1E	Voltage-dependent R-type Ca^2+^ channel subunit α-1E	
	CACNA1H	Voltage-dependent T-type Ca^2+^ channel subunit α-1 H	[11]
	CACNA1I	Voltage-dependent T-type Ca^2+^ channel subunit α-1I	[12]
	CACNA1A	Voltage-dependent P/Q-type Ca^2+^ channel subunit α-1A	
	CACNA1C	Voltage-dependent L-type Ca^2+^ channel subunit α-1C	
	CACNA1D	Voltage-dependent L-type Ca^2+^ channel subunit α-1D	[11]
	CACNA1G	Voltage-dependent T-type Ca^2+^ channel subunit α-1G	[11,12,74]
	CACNA1S	Voltage-dependent L-type Ca^2+^ channel subunit α-1S	
	ORAI1	Calcium release-activated Ca^2+^ channel protein 1	
Ca^2+^ transporters and exchange proteins in plasma membrane	SLC8A1	Na^+^ -Ca^2+^ exchanger 1	[11,12,74]
	SLC8A2	Na^+^ -Ca^2+^ exchanger 2	
	ATP2B1	Plasma membrane Ca^2+^-transporting ATPase 1 (PMCA1)	
	ATP2B2	Plasma membrane Ca^2+^-transporting ATPase 2 (PMCA2)	
	ATP2B3	Plasma membrane Ca^2+^-transporting ATPase 3 (PMCA3)	
	ATP2B4	Plasma membrane Ca^2+^-transporting ATPase 4 (PMCA4)	
Ca^2+^ channels and transporters in endoplasmic reticulum	ITPR1	Inositol 1,4,5-trisphosphate receptor type 1 (IP3R 1)	[12]
	ITPR2	Inositol 1,4,5-trisphosphate receptor type 2 (IP3R 2)	[11,12]
	ITPR3	Inositol 1,4,5-trisphosphate receptor type 3 (IP3R 3)	[33]
	STIM1	Stromal interaction molecule 1	[11,12,33]
	STIM2	Stromal interaction molecule 2	
	RYR1	Ryanodine receptor 1	
	RYR2	Ryanodine receptor 2	[11]
	RYR3	Ryanodine receptor 3	[12]
	ATP2A1	Sarcoplasmic/endoplasmic reticulum Ca^2+^ATPase 1 (SERCA1)	
	ATP2A2	Sarcoplasmic/endoplasmic reticulum Ca^2+^ATPase 2 (SERCA2)	
	ATP2A3	Sarcoplasmic/endoplasmic reticulum Ca^2+^ ATPase 3 (SERCA3)	
Ca^2+^-binding proteins	CALML3	Calmodulin-like protein 3 (CaM-like protein)	[12]
	PTK2B	Ca^2+^-dependent tyrosine kinase 2β	[12]
	S100A8, S100A9, S100A11, S100P	S100 Ca^2+^-binding protein A8, A9, A11 and P	[75]
	ITPKB	Inositol-trisphosphate 3-kinase B	[12]
	PDF1A	Ca^2+^/calmodulin-dependent 3’,5’-cyclic nucleotide phosphodiesterase 1A	
	PDE1C	Ca^2+^/calmodulin-dependent 3’,5’-cyclic nucleotide phosphodiesterase 1C	
	PDE1B	Ca^2+^/calmodulin-dependent 3’,5’-cyclic nucleotide phosphodiesterase 1B	
	PPP3CB	Calmodulin-dependent calcineurin A subunitβ isoform	
	PPP3CC	Calmodulin-dependent calcineurin A subunit γ isoform	

**Table 4 cancers-12-02755-t004:** Ca^2+^-signaling genes under or overexpressed in human HCC tissue.

Ca^2+^ Signaling Pathway	Gene	Protein	Change in Protein Expression	References
Ca^2+^ channels and transporters in plasma membrane	ORAI1	Orai 1	Increased	[27]
	CACNA1H	Voltage-operated Ca^2+^ channel subunit α-1 H	Increased	[77]
	TRPC6	Transient receptor potential cation channel subfamily C member 6	Increased	[28]
	TRPM2	Transient receptor potential cation channel subfamily M member 2	Increased	[78]
	TRPV2	Transient receptor potential cation channel subfamily V member 2	Increased	[79]
	TRPV4	Transient receptor potential cation channel subfamily V member 4	Decreased	[78]
Ca^2+^ channels and transporters in endoplasmic reticulum	STIM1	Stromal interaction molecule 1	Increased	[80]
	SERCA2	Sarco/endoplasmic reticulum (Ca^2+^, Mg^2+^) ATP-ase	Decreased (in non-alcoholic steatohepatitis-induced HCC)	[23]
Ca^2+^ channels and transporters in mitochondria	MCUR1	Mitochondrial Ca^2+^ uniporter regulator 1	Increased	[81]
Ca^2+^-binding proteins	HRC	Histidine-rich Ca^2+^-binding protein	Increased	[82]
	NCS1	Neuronal Ca^2+^ sensor 1	Increased	[83]
	CAB39	Ca^2+^-binding protein 39	Increased	[39]

**Table 5 cancers-12-02755-t005:** Potential targets involving Ca^2+^-signaling pathways for therapeutic intervention directed to prevention of development of hepatocellular carcinoma in non-alcoholic fatty liver patients ^a.^

Proposed Ca^2+^ Transporter, Channel or Ca^2+^-Binding Protein	Proposed Intervention Strategy	References
SERCA2b	Activation using small molecule activator such as the allosteric activator CDN1163	[22]
	Activation by modification of ER membrane fluidity affected by altering thioesterase superfamily member 2/phosphatidyl transfer protein	[90]
	Increased expression induced by maresin 1 leading to increased AMPK activity	[91]
	Activation by modulation of the SERCA2b regulator protein Cisd2	[23]
Ca^2+^ entry	Activation of store-operated Ca^2+^ entry by small molecule activator of Orai1 or STIM1.	[20]
	Inhibition of PKC leading to de-phosphorylation and activation of Orai1	[92]
	Inhibition of Ca^2+^ entry using Ca^2+^-channel blockers verapamil and nifedipine	[21,93]
Ryanodine receptors (RYR1 and RYR2)	Activation using small molecules such as caffeine and caffeine analogs	[11]
InsP_3_R	Inhibition using small molecule inhibitors such as heparin and caffeine	[11,94]
CaMKII	Inhibition by natural product tetrandrine of phosphorylation of CaMKII	[37]
CaMKK2	Inhibition using small molecule inhibitor such as STO-609	[25]

^a^ Taken from Ali et al. 2019 [5], with permission.

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
