# Peer review of "Targeting Ca2+ Signaling in the Initiation, Promotion and Progression of Hepatocellular Carcinoma"

_cancers, 2020, doi:10.3390/cancers12102755_

Round 1
Reviewer 1 Report
There are a number of spelling errors throughout the paper (some examples are listed below):
Line46: Hepatitis B is HBV, not HVC
Line55: HBC infection? Did you mean HCV infection? Or are you referring to HBc – hep b core antibody?
Line104: calls should be cells
Line128: 20,000 what?
Line252: loose should be lose
Line285: numerus should be numerous
Line944: CanKII should be CamKII
Line 992: ‘of’ should be ‘or’
You have mislabeled CACNA1I, CACNA1C, and CACNA1D in TABLE 3. I stopped after the third error and did not check the rest of the table.
You have Cav3.2 listed under the GENE NAME for Table 4. Should only ready CACNA1H
The organization of tables 3 and 4 are not consistent. Table 3 has 'protein name' before 'gene name' and table 4 has 'gene name' before 'protein name'.
Throughout the manuscript you (mostly in the first half) you have a number of run-on sentences as well as a number of segments that are quite difficult to discern due to their convoluted nature.
I can tell that a significant amount of time and effort was dedicated to writing this manuscript but it still has a significant number of mistakes that require correction.
Author Response
Reviewer # 1:
There are a number of spelling errors throughout the paper (some examples are listed below):
Line46: Hepatitis B is HBV, not HVC
Line55: HBC infection? Did you mean HCV infection? Or are you referring to HBc – hep b core antibody?
Line104: calls should be cells
Line128: 20,000 what?
Line252: loose should be lose
Line285: numerus should be numerous
Line944: CanKII should be CamKII
Line 992: ‘of’ should be ‘or’
The spelling issues have been corrected at the specific locations in the manuscript and we have tried to search as far as possible for others.
Line128: 20,000 what?
The 20,000 refers to a gradient, so has no units. The word gradient is mentioned, so we think it is OK to leave statement as it was.
You have mislabeled CACNA1I, CACNA1C, and CACNA1D in TABLE 3. I stopped after the third error and did not check the rest of the table.
You have Cav3.2 listed under the GENE NAME for Table 4. Should only ready CACNA1H
We have corrected Table 3 and 4, as summarized above.
The organization of tables 3 and 4 are not consistent. Table 3 has 'protein name' before 'gene name' and table 4 has 'gene name' before 'protein name'.
We have modified the columns in Tables 3 and 4 to be consistent: Ca2+ signaling pathway, Gene, Protein. The comment “Comments” column has been omitted from Table 3.
A number of other minor aspects of wording and abbreviations in the Tables have also been improved.
We have also checked the formatting of the Tables, as below.
Throughout the manuscript you (mostly in the first half) you have a number of run-on sentences as well as a number of segments that are quite difficult to discern due to their convoluted nature.
We have gone through the manuscript, and divided long run on sentences into two and tried to clarify some of the indirect statements. We hope this has improved the clarity of the writing.
Reviewer 2 Report
This manuscript provides an up-to-date review and focuses on the latest and outstanding developments on the molecular research of the specific topic, namely Ca2+ signaling proteins which may play important roles during carcinogenesis of HCC. In the manuscript, the potential therapeutic targets of Ca2+ signaling proteins are summarized and grouped based on their roles during hepato-carcinogenesis stages. These include proteins identified to be associated with liver cancer stem cells, initiation and promotion by HBV and HBC infection and non-alcoholic fatty liver disease, and in established HCC tumors, including migration and metastasis. The reviewer supports the publication of this manuscript with enthusiasm.
Minor point:
The format for all the Tables, namely Table 1 to Table 5, presented in the current manuscript is not clear enough to separate different items and looks a bit confused. Please make revision.
Author Response
Reviewer #2:
Minor point:
The format for all the Tables, namely Table 1 to Table 5, presented in the current manuscript is not clear enough to separate different items and looks a bit confused. Please make revision.
For the Tables in general we have done three overall things.
Corrected Tables 3 and 4 as above.
Made better a number of small details.
Modified as best possible the setting out of text in each Table. We believe that the MDPI Cancers submission system converts MS Word table line into “Tab/no line” leading to possible distortion of the columns. We hope that what we have done will solve this problem.
Reviewer 3 Report
In this review entitled “Targeting calcium signalling in the initiation, promotion and progression of hepatocellular carcinoma”, Ali et al. present a very comprehensive overview of HCC encompassing clinical data, screening results, potential treatments, as well as the associated calcium signalling pathways. Many schemes are provided to illustrate the main signalling pathways described in the review. Several tables are also presented, but probably due to a problem with the conversion into a pdf file, lines and columns are not apparent, making them more complicated to understand than necessary.
My only suggestion to the authors would be to remove the last sentence from the abstract (lines 30-32), since the DEB-TACE method only represents a minor point in their manuscript.
I have also listed some typos:
Line 316: PD-L1 instead of PD-1
Table 2: Severe combined immunodeficient (SCID instead of NCID) mice
Line 495: BCL2 instead of BLC2
Line 698: TRPC6 instead of TRPM6
Author Response
Reviewer #3:
Several tables are also presented, but probably due to a problem with the conversion into a pdf file, lines and columns are not apparent, making them more complicated to understand than necessary.
We have tried to address this table format problem as described above.
My only suggestion to the authors would be to remove the last sentence from the abstract (lines 30-32), since the DEB-TACE method only represents a minor point in their manuscript.
We have omitted this from the abstract. In addition, we have deleted the DEB-TACE subheading and incorporated that section into the HCC treatment subsection.
I have also listed some typos:
Line 316: PD-L1 instead of PD-1
Table 2: Severe combined immunodeficient (SCID instead of NCID) mice
Line 495: BCL2 instead of BLC2
Line 698: TRPC6 instead of TRPM6
We have corrected these at the specific locations. In addition, we have searched and corrected as best possible other typographical errors.